# Formulation and Safety Tests of a *Wickerhamomyces anomalus*–Based Product: Potential Use of Killer Toxins of a Mosquito Symbiotic Yeast to Limit Malaria Transmission

**DOI:** 10.3390/toxins13100676

**Published:** 2021-09-23

**Authors:** Alessia Cappelli, Consuelo Amantini, Federica Maggi, Guido Favia, Irene Ricci

**Affiliations:** 1School of Biosciences and Veterinary Medicine, University of Camerino, 62032 Camerino, Italy; alessia.cappelli@unicam.it (A.C.); consuelo.amantini@unicam.it (C.A.); guido.favia@unicam.it (G.F.); 2Department of Molecular Medicine, Sapienza University, 00185 Rome, Italy; federica.maggi@uniroma1.it; 3Immunopathology Laboratory, School of Pharmacy, University of Camerino, 62032 Camerino, Italy

**Keywords:** yeast killer toxin, *Wickerhamomyces anomalus*, symbiotic control, malaria, mosquitoes, vector-borne diseases, freeze drying, safety

## Abstract

*Wickerhamomyces anomalus* strain *Wa*F17.12 is a yeast with an antiplasmodial property based on the production of a killer toxin. For its symbiotic association with *Anopheles* mosquitoes, it has been proposed for the control of malaria. In an applied view, we evaluated the yeast formulation by freeze-drying *Wa*F17.12. The study was carried out by comparing yeast preparations stored at room temperature for different periods, demonstrating that lyophilization is a useful method to obtain a stable product in terms of cell growth reactivation and maintenance of the killer toxin antimicrobial activity. Moreover, cytotoxic assays on human cells were performed, showing no effects on the cell viability and the proinflammatory response. The post-formulation effectiveness of the killer toxin and the safety tests indicate that *Wa*F17.12 is a promising bioreagent able to impair the malaria parasite in vector mosquitoes.

## 1. Introduction

*Wickerhamomyces anomalus* (formerly *Pichia anomala* and *Hansenula anomala*) is a ubiquitous ascomycete yeast that is isolated from different sources, such as food, plants and insects [1]. In the last decade, this yeast has gained great attention for its applicability in food biopreservation and potential use in medicine, thanks to its ability to produce killer toxins (KTs) with a wide antimicrobial activity [1]. *W. anomalus* is deemed by the European Food Safety Authority (EFSA) to be a microbiological agent with a Qualified Presumption of Safety (QPS) status for use in the food industry [2]. Nevertheless, in-depth studies of the killer yeast effects on humans and the environment as well as tests of the maintenance of the KT-based antimicrobial properties in formulated products are necessary for its further biotechnological applications.

Interestingly, for its association as a mutualistic symbiont with insects that transmit diseases to humans and animals, such as mosquitoes and sandy flies, *W. anomalus* has been proposed also for the symbiotic control (SC) of vector-borne diseases (VBDs) [3,4]. Malaria is one of the most troubling VBDs, and the fight against vector mosquitoes is mainly based on insecticides. On the other hand, the insurgence of insecticide resistance limits the efficacy of this method, and thus, innovative strategies are urgently needed [5]. Taking into account also the toxicity and high costs of insecticides, the development of alternative strategies by the use of sustainable and eco-friendly products are under investigation [6]. A possible approach involves the dissemination in vector mosquitoes of inherited symbiotic yeasts that kill the malaria pathogen (*Plasmodium*) from an infected blood meal, by releasing KTs in the female midgut [3]. This strategy implicates the release of killer symbionts in malaria endemic areas to reduce the vector capability and block the transmission. Several studies showed that the strain of *W. anomalus Wa*F17.12, which localizes in the mosquito midgut and gonads and is vertically transmitted, impairs *Plasmodium berghei* in *Anopheles stephensi* (the major Asian malaria vector) through a KT-mediated effect that causes parasite membrane damage and death [7,8].

The antiplasmodial effect of the KT secreted by *Wa*F17.12 (*Wa*F17.12-KT) and its biological relationship with the female mosquito are pivotal features for the SC of malaria. Nevertheless, the implementation of such a killer yeast–based product and the development of a commercialization pathway require additional investigations that include three main objectives: (i) assessment of effective delivery systems (e.g., dissemination in vector insects), (ii) optimization of low-cost industrial-scale culturing and storing (e.g., lyophilization), and (iii) evaluation of the toxicity. Concerning yeast dissemination, Cappelli and collaborators have demonstrated that the sugar diet in laboratory-reared mosquitoes can be supplemented with *Wa*F17.12, maintaining a KT-mediated antiplasmodial effect until the next generation [8]. In an applied view, the *Saccharomycetes* yeast could be dispersed on flowers, and the volatile substances produced by nectar fermentation would attract mosquitoes to their food sources [9,10]. Even though this strategy seems applicable, the question of how to turn yeast into a product remains open and requires demonstrations that it is cultivable on a large scale and formulable, i.e., resistant to transformation processes in terms of viability and biocontrol activity maintenance, as well as being absolutely safe. A lyophilized yeast preparation could be stored at room temperature (rt) and easily released in the environment at low cost.

In the present work, we performed formulation and safety tests, focusing on *Wa*F17.12. The effect on the growth reactivation and the maintenance of the killer toxin activity after freeze-drying was evaluated, demonstrating that *Wa*F17.12 is a stress-tolerant strain that is transformable into a dried product, ready to use in the field. Concerning the safety, despite several killer strains of *W. anomalus* being used in the food industry [11], there are only a few studies demonstrating that mice orally treated with the yeast did not show collateral effects [12] and KTs did not harm murine hepatocytes and human erythrocytes in vitro [8,13]. Here, we estimated the effect of *Wa*F17.12 in terms of cell viability and the proinflammatory response on human keratinocyte cell lines, which represent the first barrier of the human body. The overall results obtained in this study represent a step forward in the commercialization of a KT producer strain of *W. anomalus* associated with vector mosquitoes as an innovative tool to prevent the spreading of VBDs.

## 2. Results

### 2.1. Growth Reactivation and Killer Toxin Activity Maintenance of WaF17.12 after Lyophilization

Lyophilization is one of the most successful and convenient methods to preserve high cell viability; thus, it is a common industrial technique to preserve microorganisms for a long time [14]. Nevertheless, the freeze-drying process could cause cell damage by ice crystals formation and an intracellular increase in the salt concentration and water leak, and thus, not all microbes are suitable for this treatment [15]. To investigate the possible use of freeze-drying of *W. anomalus*, the yeast growth rate and the KT activity of lyophilized *Wa*F17.12 were analyzed after different periods of storage (rt) (4, 12, 18, 25, 32 and 60 days). Analysis of the yeast cultures was performed after 36 h of incubation at optimal conditions for stimulating the secretion of *Wa*F17.12-KT [16] (Figure 1).

The cell counts were estimated using the trypan blue assay, which allowed to distinguish viable cells since staining occurs only in dead or damaged cells (Figure 1A). The growth rate showed no differences of lyophilized *Wa*F17.12 versus the fresh sample (Ctr) up to 32 days of storage, while a slowdown was detected after 60 days (*p* < 0.05). The results demonstrated constant yeast viability in terms of cell replication up to the fourth/fifth week post-lyophilization. Moreover, the trypan blue assay indicated that the yeast conserved cell membrane integrity in all the analyzed storage periods. This suggests that after two months of storage, the observed lower cell concentration is likely due to a slower rate of cell replication rather than a greater number of damaged or dead cells.

To evaluate the maintenance of the *Wa*F17.12-KT activity after lyophilization, the antimicrobial effect of supernatants from the previously analyzed yeast cultures was tested against the sensible strain *Wa*UM3 (Figure 1B). This analysis was performed because *Wa*F17.12-KT is a soluble glycoprotein that is secreted under optimal growing conditions, and its presence in the supernatants was checked by Western blot, using a monoclonal antibody as previously described by Cappelli et al. [16]. *Wa*F17.12-KT showed an unaltered antimicrobial activity against *Wa*UM3, even in supernatants of cultures from yeast reactivated 60 days post-lyophilization likely due to the reaching of a secretion plateau at a cell concentration of 10^7^–10^8^. The results suggested that, despite a slowdown in the yeast replacement by the longest period, the ability of *Wa*F17.12 to maintain a killer phenotype remains unchanged.

### 2.2. Effects of WaF17.12 on Viability of Human Keratinocytes (HaCaT)

The effects of different concentrations of *Wa*F17.12 were tested on human keratinocytes since they represent the first defensive barrier of the body and are a useful target for cytotoxic studies [17]. The HaCaT cell line, a non-tumorigenic monoclonal spontaneously immortalized cell line that exhibits normal morphogenesis and expresses all the major surface markers and functional activities of isolated keratinocytes, was used as a model [18]. Cells were treated with 1000, 5000 and 10,000 yeast cells/mL, and the treatment effects on the viability of keratinocytes were evaluated at 24 h and 48 h (Figure 2).

No morphological alterations, no cell detachment and no changes at the differentiation level were found by light microscopy in HaCaT cells co-cultured with different concentrations of *Wa*F17.12 (Figure 2A). Interestingly, the yeast does not adhere to the surface of the cell layer and grows suspended in the HaCaT medium as demonstrated by the yeast removal through washing (Figure 2AI,II).

To deeply investigate the effects induced by *Wa*F17.12 on HaCaT viability, MTT assay, and propidium iodide (PI) staining and FACS analysis were performed. The MTT assay detected no decrease in the number of viable cells in the samples co-cultured with yeasts (non-treated HaCaT cells were used as control =100%) (Figure 2B). The PI assay supported the results by MTT, demonstrating no increment in the cell death rate (MFI = mean fluorescence intensity) (Figure 2C). The overall results showed that treatments with different concentrations of *Wa*F17.12 for 24 h and 48 h did not have cytotoxic effects on human keratinocytes and did not affect cell viability. 

### 2.3. Effect of WaF17.12 on the Proinflammatory Response

Keratinocytes are able to release several proinflammatory mediators, such as IL-1, IL-6 and TNFα, by participating directly in the immune response [19]. Moreover, in inflamed skin, the STAT3 signaling pathway was found to be activated and overexpressed [20]. Thus, we investigated whether the presence of *Wa*F17.12 in the HaCaT cell culture was able to stimulate a proinflammatory response. To this purpose, the gene expression of IL-1, IL-6, TNFα and STAT3 was assessed by real-time PCR (Figure 3). 

As shown, the incubation for 24 h or 48 h of HaCaT cells with different concentrations of *Wa*F17.12 did not increase gene expression levels in any of the four targets (non-treated HaCaT cells cultivated in standard medium were used as control). Different trends in gene expression among yeast amounts were not statistically significant (one-way ANOVA test and the Bonferroni post hoc test).

The expression of the most common inflammatory cytokine TNFα was also assessed at the protein level by Western blot analysis. Our results showed no differences in the expression of the TNFα protein in HaCaT cells co-treated with *Wa*F17.12, compared with control cells (Figure 4).

Overall, these results indicated no induction of inflammation in HaCaT cells induced by the presence of *Wa*F17.12.

## 3. Discussion

*W. anomalus* secretes extracellular KTs displaying a wide spectrum of antimicrobial activity against sensitive microorganisms that ensure a dominant role in the competition within the environmental niches. This capability has stimulated its use as a biocontrol agent in the agro-food industry to block the proliferation of undesired microorganisms [21]. Diverse environmental strains are notified to the EFSA as microbiological agents at QPS-1 and have been successfully employed against detrimental microbes, such as *Penicillium* sp. and *Botrytis cinerea,* reducing the use of chemical agents [22,23].

A new application of *W. anomalus* has been proposed in the SC of VBDs, after the discovery of strains in mutualistic association with vector insects, such as *Anopheles*, *Aedes* and *Culex* mosquitoes and *Phlebotomus perniciosus* sand fly [4,24,25]. Diverse studies have been carried out in *Anopheles* where the killer activity of the symbiotic strain *Wa*F17.12 has been demonstrated to cause structural damage and death of the malaria parasite *P. berghei* [8,26]. Cappelli and collaborators showed that *Wa*F17.12, administered through the diet, is able to stably colonize the female mosquito midgut and gonads, being vertically transmitted to the progeny [16]. Thus, the malaria parasite can be impaired in the midgut by pre-feeding mosquitoes with the killer yeast before the ingestion of an infected blood meal [8]. Since mosquitoes search the sugar for an energy supply soon after the emergence, they would be attracted by odorant feeding stations containing, for example, honey or fruit juice plus *Wa*F17.12. In this view, the dissemination of the killer yeast in newly emerged mosquitoes can be favored through the release of feeding stations near the larval breeding sites (e.g., puddles).

The biological interactions of *Wa*F17.12 with the vector and the parasite guarantee the potential success of this strategy. Nevertheless, a yeast delivery system should be implemented through an inexpensive, environmentally friendly and safe product. In applied view, a lyophilized yeast preparation ready-to-use can be included in the packaging of feeding stations. Several strains of *W. anomalus* used as biocontrol agents in the food industry produce biomass on a large scale [27] and exert stress tolerance after dehydration [15]. On this background, we tested the freeze-drying of *Wa*F17.12, demonstrating yeast survival and the killer toxin activity maintenance up to two months of rt storage, and proposing transformation processes that do not require high-cost procedures, such as low temperatures or vacuum systems.

Concerning the *W. anomalus* safety aspect, the evaluation potential toxicity plays a critical role, although the yeast is considered to be non-virulent and is reported rarely as an opportunistic pathogen in immunocompromised patients [28]. On the other hand, a large screening of donors, including different classes of patients and healthy people exposed to mosquito bites, detected a single *W. anomalus* fungemia positive case in an immunocompromised patient [29]. In fact, the biochemical characterization of *Wa*F17.12-KT revealed a glycoproteic structure with an exo-β-1,3-glucanase activity that binds to specific receptors on the surface of microbes but does not target mammalian cells [30]. To our knowledge, there are a few studies in vitro or in vivo on the toxicity of *W. anomalus* and KTs [8,12,13]. Although yeast KTs appear to be harmless to non-microbial targets, it is important to establish that there are no other yeast toxicity mechanisms. Therefore, in the applied view of SC strategies, we have carried out tests, using live yeast. Our investigations contributed to fill this gap of knowledge, demonstrating that the treatment with *Wa*F17.12 did not affect the cell viability and did not stimulate the production of inflammatory factors in human keratinocytes, which represent the major cell type of the epidermis. 

This study suggests the use of the symbiotic killer yeast *Wa*F17.12 as a bioreagent for killing the malaria parasite in vector mosquitoes and limiting the malaria transmission through a sustainable and safe approach.

## 4. Conclusions

Formulation and safety tests performed in the present study showed that *Wa*F17.12 is transformable into a dried product ready to use in the field. The overall data suggest the use of symbiotic killer yeasts as sustainable and safe bioreagents for controlling the malaria spreading.

## 5. Materials and Methods

### 5.1. Yeast

The KT producer strain of *W. anomalus Wa*F17.12, isolated from *An. stephensi* mosquitoes [7,16], was tested in this study. The KT non-producer strain *Wa*UM3 was used as a target system, susceptible to *Wa*F17.12-KT, in the antimicrobial activity assay [16,31]. For all the experiments of the present study (post-lyophilization assays and tests on keratinocytes), *Wa*F17.12 was cultured at the optimal growth condition for stimulating the KT production; cells were incubated at 26 °C and 70 rpm for 36 h in YPD broth (20 g/L peptone, 20 g/L glucose, 10 g/L yeast extract), buffered at 4.5 pH with 0.1 M citric acid and 0.2 M K_2_HPO_4_ [16]. 

### 5.2. HaCaT Cells Cultures

An immortalized human keratinocyte (HaCaT) cell line (Creative Bioarray, Shirley, NY, U.S.A.) was cultured in DMEM (Lonza, Allendale, NJ, U.S.A.) supplemented with 10% fetal bovine serum (FBS) (EuroClone, Milano, Italy), 2 mM L-glutamine (Lonza, Allendale, NJ, U.S.A.), 100 IU/mL of penicillin, 100 µg of streptomycin (Lonza, Allendale, NJ, U.S.A.) and maintained at 37 °C with 5% CO_2_ and 95% humidity. 

### 5.3. Freeze-Drying of WaF17.12

After cultivation, the WaF17.12 cells were counted with trypan blue in the Neubauer chamber. After two washings with 0.9% NaCl solution, six vials containing 10^7^ cells/mL in fresh medium (YPD pH 4.5) were prepared, and 5% trehalose was added to preserve the cell viability in the dry state [32]. The samples were immediately frozen (at −80 °C) for 12 h; then, the cultures were transferred to a freeze-dryer (Edwards, Burgess Hill, United Kingdom). Freeze-drying was carried out for 24 h at −50 °C and at a pressure of 0.15 mbar [33]. The vials containing lyophilized strains were sealed with Parafilm (Darmstadt, Germany) and stored up to 60 days at rt and protected from light.

### 5.4. Growth Rate Assessment and Killer Toxin Activity Assay Post-Lyophilization

The viable cell number and the antimicrobial activity of WaF17.12-KT were checked in the different yeast preparations (4, 12, 18, 25, 32 and 60 days post-lyophilization). Per each preparation, 10^7^ lyophilized cells/mL were rehydrated in sterile water and incubated for culturing, whereas 10^7^ fresh (non-lyophilized) cells/mL of *Wa*F17.12 were used as control. All yeast samples were cultured at optimal growth conditions for the KT production (see above). After incubation, the *Wa*F17.12 cells were counted by trypan blue in the Neubauer chamber to assess the yeast growth rate after different days of storage. For the killer toxin activity assay, the cultures from the lyophilized samples and the control were centrifuged at 3000 rpm for 5 min, and the supernatant was filtered to eliminate cells. The presence of *Wa*F17.12-KT was detected in all the supernatant by Western blot, using the monoclonal antibody mAbKT4 [16,31]. To test the maintenance of *Wa*F17.12-KT activity after lyophilization, 10^6^ cells of *Wa*UM3 were incubated overnight at 26 °C and 70 rpm in YPD pH 4.5 diluted with each supernatant (1:1). Since supernatants deriving from previous cultures are to be considered a growth medium that is partially exhausted, the control was performed, using 10^6^ cells of *Wa*UM3 incubated in YPD pH 4.5 diluted with a self-supernatant of 36 h (1:1). After incubation, *Wa*UM3 growth was evaluated by counting with trypan blue in the Neubauer chamber. All samples were assayed in triplicate and the experiment was performed twice. The mean ± SEM for each group was considered. Statistical analyses were performed using GraphPad Prism 5 software. Multiple comparisons using Mann–Whitney test determined the statistical difference in *Wa*F17.12 growth at different time points. The KT activity against *Wa*UM3 was evaluated in the six treated groups, using the two-way ANOVA and Bonferroni post hoc test. A *p*-value < 0.05 was considered statistically significant.

### 5.5. Treatment of HaCaT Cells with WaF17.12

The 2 × 10^5^ cells/well were plated in 6-well plates and incubated overnight in supplemented DMEM at 37 °C, 5% CO_2_ and 95% of humidity. The cells were treated for 24 h and 48 h with different concentrations of *Wa*F17.12: 1000, 5000 and 10,000 yeasts cells/mL. Untreated cells were used as the control. The stimulation was performed in triplicate experiments. The morphology of keratinocytes was analyzed by bright field microscopy, using a fluorescence microscope (Olympus, Milan, Italy).

### 5.6. Cell Viability Assays in HaCaT

Cell death and/or viability induced by different *Wa*F17.12 concentrations (1000, 5000, 10,000 yeasts cells/mL) after 24 h and 48 h was evaluated, using MTT (3-(4,5-dimethylthiazol-2-yl)-2,5-diphenyltetrazolium bromide) and PI (Sigma-Aldrich, Saint Louis, U.S.A.) assays. 

In MTT assay, HaCaT cells (7.5 × 10^5^ cells/mL) were seeded into 96-well plates and co-cultured with different yeast concentrations or the vehicle (DMEM). Upon treatment, the supernatant was removed, and the cells were washed twice with PBS 1×. MTT (0.8 mg/mL) was added to the samples and incubated for 3 h. Then, the supernatants were discarded, and 100 μL/well DMSO were added to dissolve the formazan crystals. The colored solutions were read by a microtiter plate spectrophotometer (BioTek Instruments, Winooski, VT, U.S.A.). Six replicates were used for each treatment. The experiments were repeated three times.

For the PI assay, HaCaT cells were incubated with 20 µg/mL of PI in 1× PBS, for 30 min at 20 °C. PI penetrates cells with altered membranes intercalating with the broken DNA, a typical process of cell necrosis. Samples were analyzed by a FACScan cytofluorimeter, using the CellQuest software. The experiments were repeated three times. 

### 5.7. Gene Expression Analysis of Proinflammatory Mediators

The relative expression of four genes involved in the pro-inflammatory response (IL-1, IL-6, TNFα and STAT-3) was evaluated. HaCaT cells were treated with *Wa*F17.12 as previously described. Total RNA was extracted with RNAzol^®^ RT following the manufacturer instructions (Sigma-Aldrich, St. Louis, MI, U.S.A.). Total RNA (1 µg) was subjected to reverse transcription, using PrimeScript™ RT reagent Kit (Takara, Kusatsu, Giappone), and qRT-PCR was performed by using CFX96 Touch Real-Time PCR Detection System (BioRad, Hercules, CA, U.S.A.). The reaction mixture contained 1× PCR Brilliant Multiplex QPCR Master Mix (Agilent, Stratagene), PrimePCR Probe Assays with FAM or HEX fluorophore (BioRad, Hercules, CA, U.S.A.) used, according to company datasheet. PCR parameters were 10 min at 95 °C followed by 40 cycles of 95 °C for 30 s and 55 °C for 30 s and 72 °C 30 s. All samples were assayed in triplicate. The relative amount of target mRNA of IL-1, IL-6, TNFα, STAT3 and GAPDH (housekeeping gene) was calculated by the 2-ΔΔCt method. Statistical analysis was performed using the Bio-Rad CFX Manager Software and the GraphPad software (http://www.graphpad.com). For each group, mean values and the standard error (SEM) from two independent experiments were calculated. One-way ANOVA test and the Bonferroni post hoc test were used to assess the statistical differences between gene expressions.

### 5.8. Western Blot Analysis of TNFα 

HaCaT cells untreated or treated with different *Wa*F7.12 concentrations (1000, 5000 and 10,000 yeasts cells/mL) for 24 h and 48 h were suspended in a lysis buffer containing protease inhibitor cocktail (Sigma Aldrich, St. Louis, MO, U.S.A.). Lysates were separated on 14% SDS polyacrylamide gel in a Mini-PROTEAN Tetra Cell system (BioRad, Hercules, CA, U.S.A.) and transferred onto nitrocellulose membranes (EuroClone, Milano, Italy), using Mini Trans-Blot Turbo RTA system (BioRad). Blots were incubated with a blocking solution containing 3% bovine serum albumin (BSA) in PBS 1× and 0.1% Tween 20 for 1 h at rt. Membranes were incubated overnight at 4 °C in anti-human TNFα primary antibody (Ab) diluted 1:1000 in PBS tween + BSA 3% (Enzo Life Science, Farmingdale, New York, U.S.A.) followed by incubation for 1 h at rt with HRP-conjugated anti-mouse secondary Ab. As a loading control, membranes were incubated with anti-GAPDH 1:1000 PBS-tween in milk 5% for 1 h at rt followed by the appropriate HRP-conjugated secondary Ab, according to the specific datasheet. Detection was performed, using the LiteAblot ^®^PLUS kit (EuroClone, Milano, Italy). As positive control for the TNFα expression, HaCaT cells were treated with 1 µM etoposide for 24 h [17].

## Figures and Tables

**Figure 1 toxins-13-00676-f001:**
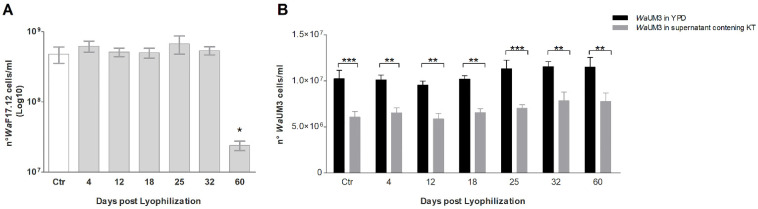
Analysis of the *Wa*F17.12 growth rate and KT activity against the sensible target *Wa*UM3 after. (**A**) Growth rate of *Wa*F17.12 cultures was estimated at 4, 12, 18, 25, 32 and 60 days of room temperature storage post freeze-drying, Ctr: non-lyophilized *Wa*F17.12 culture. Cell counts were carried out using trypan blue assay. (**B**) Antimicrobial activity of supernatants containing *Wa*F17.12-KT; Black bars: *Wa*UM3 cultured overnight in YPD pH 4.5 (control); Grey bars: *Wa*UM3 cultured in YPD pH 4.5 diluted with supernatants containing *Wa*F17.12-KT (1:1). Cell counts were carried out using trypan blue assay. Both the experiments were performed twice, and the bars represented the mean ± SEM for each group (* *p* < 0.05; ** *p* < 0.01; *** *p* < 0.001).

**Figure 2 toxins-13-00676-f002:**
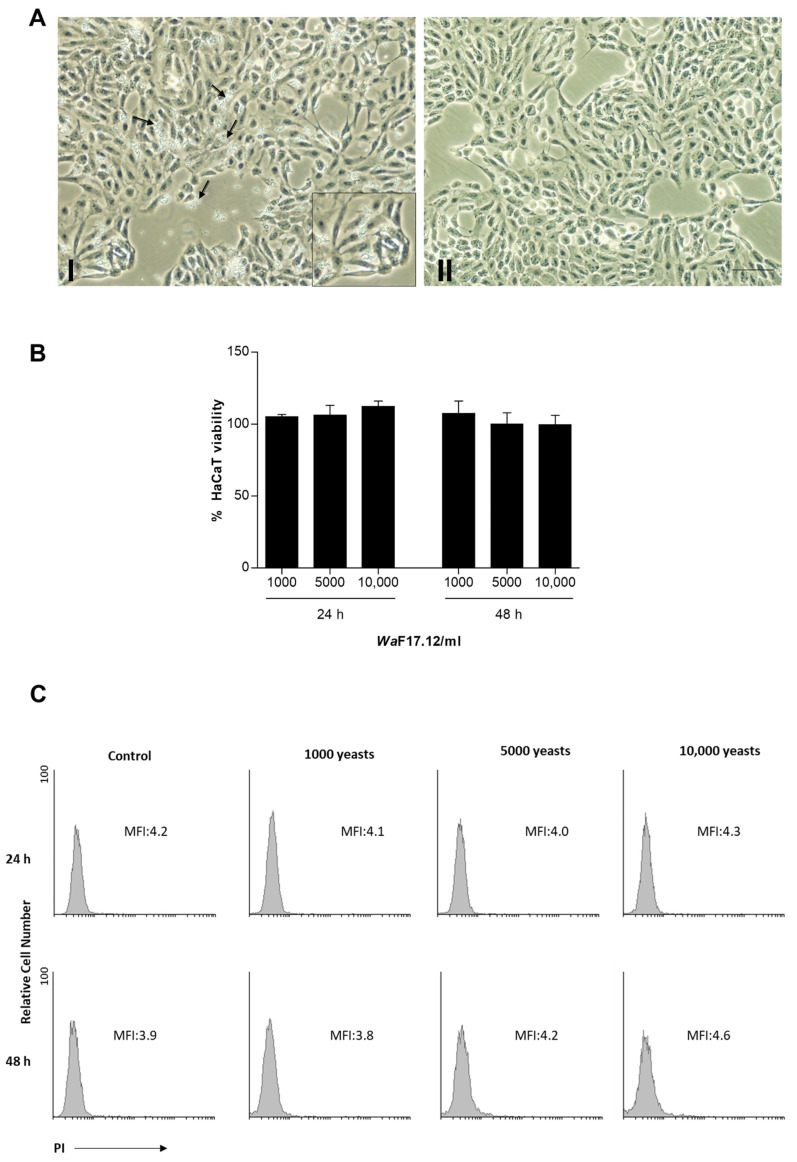
(**A**) Light microscopy of HaCaT treated with *Wa*F17.12 (10,000 yeast cells/mL) for 48 h: cells before (I) and after (II) washing. The arrows indicate yeasts onto the cell layer. In the magnification, a detail of the yeasts. Bar = 200 μm. One representative out of three independent experiments is shown. (**B**) MTT assay was performed in HaCaT cells co-cultured with different concentrations of *Wa*F17.12 (1000, 5000 and 10,000 yeasts cells/mL) for 24 h and 48 h. Data are the mean ± SD of three separate experiments. HaCaT cells in the medium were used as control (=100%). (**C**) PI staining and FACS analysis were performed in HaCaT cells. One representative out of three independent experiments is shown. MFI = mean fluorescence intensity.

**Figure 3 toxins-13-00676-f003:**
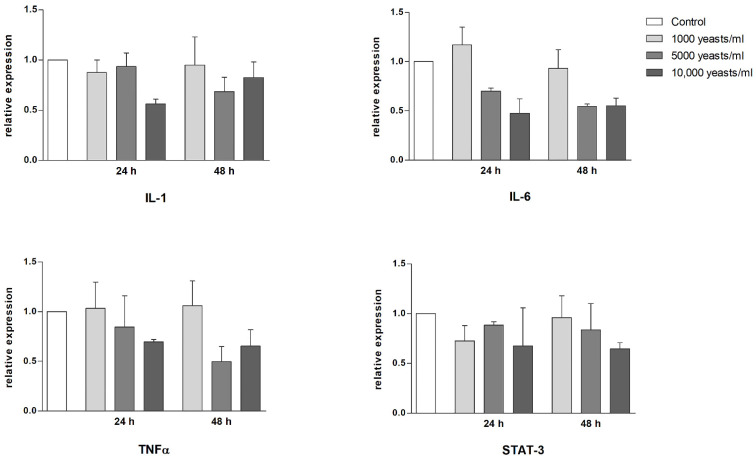
Transcriptional modulation of proinflammatory cytokines (IL-1, IL-6 and TNFα) and STAT3 genes in HaCaT cells, co-cultured in presence of different concentrations of *Wa*F17.12 for 24 h and 48 h or with standard medium (control), evaluated by quantitative RT-PCR. Data are the mean ± SEM of triplicate samples from two separate experiments and the results are normalized for GAPDH expression used as a housekeeping gene. mRNA levels were expressed as relative fold with respect to HaCaT cells in medium used as control (=1).

**Figure 4 toxins-13-00676-f004:**
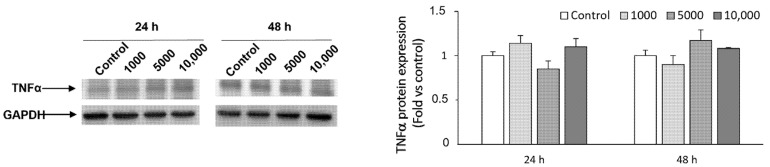
TNFα protein expression was assessed by Western blot analysis in HaCaT cells, co-cultured with different concentrations of *Wa*F17.12 or medium (control) for 24 h and 48 h. GAPDH was used as loading control. Immunoblots are representative of three separate experiments. Data of densitometric analysis are the mean ± SD of three separate experiments. Fold represents changes in protein expression with respect to control (=1).

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
