# Peer review of "Formulation and Safety Tests of a Wickerhamomyces anomalus–Based Product: Potential Use of Killer Toxins of a Mosquito Symbiotic Yeast to Limit Malaria Transmission"

_toxins, 2021, doi:10.3390/toxins13100676_

Round 1
Reviewer 1 Report
This work evaluated the yeast formulation of freeze-drying, by comparing the activity (e.g., cell growth reactivation, maintenance of the killer toxin antimicrobial activity, cytotoxicity on human cells, and the proinflammatory response) of yeast WaF17.12 after lyophilization for different periods. The topic is interesting and the paper is well designed. The manuscript is publishable on Toxins after revision. Here are some comments.
- The title seems quite irrelevant with the topic. The abstract should be simplified too.
- Most of the images (especially Fig. 1) are quite hard to recognize due to low resolution.
- As the killer toxin antimicrobial activity is a most important property of the agent, why didn’t the authors give more examples to test the antibacterial ability, in addition to the present WaUM3 in Fig. 1B?
The authors mentioned that, “the trypan blue assay indicated that the yeast conserved cell membrane integrity in all the analyzed storage periods”. Is there any data support?
Besides, Fig. 1A shows obvious slowdown for the agent after 60 day storage while Fig. 1B shows no much difference among those after different storage times. Why?
- In Fig. 1 and 3, “the experiments were performed twice”. Is this reasonable? Specifically, some of the error bars in Fig. 3 are huge.
- In Fig. 3, the authors perform tests with different yeast amounts (e.g., 1000, 5000, 10000 yeasts/ml) which show different performances. Some show positive dependence while some show negative. What is the reason of such an experimental design and what do these phenomena mean?
Author Response
Response to Reviewer 1 Comments
Point 1. This work evaluated the yeast formulation of freeze-drying, by comparing the activity (e.g., cell growth reactivation, maintenance of the killer toxin antimicrobial activity, cytotoxicity on human cells, and the proinflammatory response) of yeast WaF17.12 after lyophilization for different periods. The topic is interesting and the paper is well designed. The manuscript is publishable on Toxins after revision. Here are some comments.
Response 1. We thank Rev.1 for the comments and suggestions.
Point 2. The title seems quite irrelevant with the topic. The abstract should be simplified too.
Response 2. We modified the title and the abstract accordingly.
Point 3. Most of the images (especially Fig. 1) are quite hard to recognize due to low resolution.
Response 3. We added in the manuscript the high-resolution files of figures 1, 2, 3, 4.
Point 4. As the killer toxin antimicrobial activity is a most important property of the agent, why didn’t the authors give more examples to test the antibacterial ability, in addition to the present WaUM3 in Fig. 1B?
Response 4. The aim of the experiment in Figure 1B was testing the exo-β-1,3-glucanase activity of WaF17.12-KT secreted in supernatants. WaUM3 is the target system for this kind of analysis (ref: Polonelli et al., Clin Diagn Lab Immunol 1997; Cappelli et al., PlosOne 2014: Cecarini et al., Toxins 2019). We did not perform tests on bacterial targets since antibacterial properties of W. anomalus killer strains seem to be based on different mechanisms, not necessarily related to KTs.
Point 5. The authors mentioned that, “the trypan blue assay indicated that the yeast conserved cell membrane integrity in all the analyzed storage periods”. Is there any data support?
Response 5. The Trypan Blue assay is a dye exclusion test which is used to determine the number of viable cells present in a cell suspension. It is based on the principle that live cells possess intact cell membranes that exclude the dye (Strober, 2015 Cur Protoc Immunol). The cell numbers in all the analysed points in Fig.1A referred to intact cells (only a few dyed cells were detected), thus the observed lower cell concentration at 60 days is likely due to a slower cell replication rather than to an increase of damaged or dead cells.
Point 6. Besides, Fig. 1A shows obvious slowdown for the agent after 60 day storage while Fig. 1B shows no much difference among those after different storage times. Why?
Response 6. We suggest that the amount of WaF17.12-KT reaches a secretion plateau when cell concentration is between 107-108 (as in the sample 60 days). Thus, the antimicrobial activity against WaUM3 did not show a difference after different storage times. We added a sentence at pag.3 paragraph 2.1 according to the rev observation.
Point 7. In Fig. 1 and 3, “the experiments were performed twice”. Is this reasonable? Specifically, some of the error bars in Fig. 3 are huge.
Response 7. For both assays we analysed the samples in triplicate in two independent experiments, thus the number of replicates corresponds to n = 6. This was reported for fig.3 at pag. 9 par. 5.7. In accordance with the observation, we added the detail in the fig.3 legend and in material and methods for fig. 1 (par.5.4 at page. 8).
Point 8. In Fig. 3, the authors perform tests with different yeast amounts (e.g., 1000, 5000, 10000 yeasts/ml) which show different performances. Some show positive dependence while some show negative. What is the reason of such an experimental design and what do these phenomena mean?
Response 8. The aim of the experimental design was to evaluate interactions between WaF17.12 and the skin by mimicking several possible conditions that can occur, such as different amounts of yeast (low, moderate or high). Greater quantities of yeast were not experimentally possible as the yeast creates a layer that prevents cell growth. The statistical analysis indicated that the different trends in gene expression of HACAT cells exposed to different yeast amounts were not statistically significant (One-way ANOVA test and the Bonferroni post-hoc test). We added this sentence in the text at pag.6 par. 2.3 according to the rev.
Reviewer 2 Report
The submitted manuscript focuses on the properties of the strain Wickerhamomyces anomalus WaF17.12 which was found being associated, as a mutualistic symbiont, with mosquitoes that transmit malaria and, moreover, it has been recognized as a killer yeast against the pathogen Plasmodium which causes malaria. The authors studied: a) viability and killer toxin activity of the strain WaF17.12 which was rehydrated in different periods after lyophilization and storage at room temperature and b) the effect of three different concentrations of the strain WaF17.12 on the viability and proinflammatory response of human keratinocyte cell lines, which represent the first barrier of the human body. Based on the results obtained, the authors recognized the strain as a promising biological agent against the malaria parasite in vector mosquitoes.
General comment: The topic is an important subject. The manuscript is well written, clearly stated and well-organized. I recommend this manuscript for publication after minor revisions as follows:
“Surnatan”, probably supernatant, please correct this mistake throughout all document as well as in Fig. 1.
Please improve the quality of Fig. 1, it is not sharp enough.
Page 2, second paragraph: “In applied view, the saccharomycetes (please correct Saccharomyces with a capital letter and in italic) could be released (do you mean dispersed or applied?) on flowers…
Page 3, paragraph 2.2., line 6: “Cells were treated with 1000, 5000 and 10000 yeasts/well”… I recommend to use xx yeast cells/well
Please re-write the form “two hundred thousand cells/well” (Paragraph 5.5. Treatment of HaCaT cells with WaF17.12) to the form 2.105 cells/ml (as you used in paragraph 5.6)
Fig. 3. Please use a small letter for hours (h). I recommend to use control instead of medium in the figure legend.
Author Response
Response to Reviewer 2 Comments
Point 1. The submitted manuscript focuses on the properties of the strain Wickerhamomyces anomalus WaF17.12 which was found being associated, as a mutualistic symbiont, with mosquitoes that transmit malaria and, moreover, it has been recognized as a killer yeast against the pathogen Plasmodium which causes malaria. The authors studied: a) viability and killer toxin activity of the strain WaF17.12 which was rehydrated in different periods after lyophilization and storage at room temperature and b) the effect of three different concentrations of the strain WaF17.12 on the viability and proinflammatory response of human keratinocyte cell lines, which represent the first barrier of the human body. Based on the results obtained, the authors recognized the strain as a promising biological agent against the malaria parasite in vector mosquitoes.
General comment: The topic is an important subject. The manuscript is well written, clearly stated and well-organized. I recommend this manuscript for publication after minor revisions as follows:
Response 1. We thank Rev.2 for comments and suggestions.
Point 2. “Surnatan”, probably supernatant, please correct this mistake throughout all document as well as in Fig. 1.
Response 2. We modified the text accordingly.
Point 3. Please improve the quality of Fig. 1, it is not sharp enough.
Response 3. We added a high-resolution file of fig.1
Point 4. Page 2, second paragraph: “In applied view, the saccharomycetes (please correct Saccharomyces with a capital letter and in italic) could be released (do you mean dispersed or applied?) on flowers…
Response 4. We modified the text accordingly.
Point 5. Page 3, paragraph 2.2., line 6: “Cells were treated with 1000, 5000 and 10000 yeasts/well”… I recommend to use xx yeast cells/well
Response 5. We modified the text accordingly.
Point 6. Please re-write the form “two hundred thousand cells/well” (Paragraph 5.5. Treatment of HaCaT cells with WaF17.12) to the form 2.105 cells/ml (as you used in paragraph 5.6)
Response 6. We modified the text accordingly.
Point 7. Fig. 3. Please use a small letter for hours (h). I recommend to use control instead of medium in the figure legend.
Response 7. We modified Fig. 3 accordingly.
Round 2
Reviewer 1 Report
I have to say that the responses and revisions are barely acceptable. It is publishable now.